# Leveraging Digital Technology to Support Pregnant and Early Parenting Women in Recovery from Addictive Substances: A Scoping Review

**DOI:** 10.3390/ijerph20054457

**Published:** 2023-03-02

**Authors:** Phyllis Raynor, Cynthia Corbett, Delia West, D’Arion Johnston, Kacey Eichelberger, Alain Litwin, Constance Guille, Ron Prinz

**Affiliations:** 1College of Nursing, Advancing Chronic Care Outcomes through Research and iNnovation (ACORN) Center, University of South Carolina, Columbia, SC 29208, USA; 2Arnold School of Public Health, University of South Carolina, Columbia, SC 29208, USA; 3College of Education, University of South Carolina, Columbia, SC 29208, USA; 4Prisma Health Upstate, University of South Carolina School of Medicine, Greenville, SC 29605, USA; 5School of Health Research, Clemson University, Greenville, SC 29601, USA; 6College of Medicine, Medical University of South Carolina, Charleston, SC 29425, USA; 7Psychology Department, College of Arts and Sciences, University of South Carolina, Columbia, SC 29208, USA

**Keywords:** pregnant, parenting women, women, substance use, alcohol use, smoking, digital interventions, technology, telehealth, intervention

## Abstract

Little is known about digital health interventions used to support treatment for pregnant and early parenting women (PEPW) with substance use disorders (SUD). Methods: Guided by the Arksey and O’Malley’s Scoping Review Framework, empirical studies were identified within the CINAHL, PsycInfo, PubMed, and ProQuest databases using subject headings and free-text keywords. Studies were selected based on a priori inclusion/exclusion criteria, and data extraction and descriptive analysis were performed. Results: A total of 27 original studies and 30 articles were included. Varying study designs were used, including several feasibility and acceptability studies. However, efficacious findings on abstinence and other clinically important outcomes were reported in several studies. Most studies focused on digital interventions for pregnant women (89.7%), suggesting a dearth of research on how digital technologies may support early parenting women with SUD. No studies included PEPW family members or involved PEPW women in the intervention design. Conclusions: The science of digital interventions to support treatment for PEPW is in an early stage, but feasibility and efficacy results are promising. Future research should explore community-based participatory partnerships with PEPW to develop or tailor digital interventions and include family or external support systems to engage in the intervention alongside PEPW.

## 1. Introduction

Digital interventions are used more routinely to support treatment adherence and self-management for individuals with substance use disorders (SUD) [1,2], with even greater utilization during the COVID-19 pandemic [3]. Digital interventions refer to communication technologies used in the management of illness to promote overall health and wellbeing [4]. In prior studies, a wide assortment of Internet-based, text messaging, and mobile application interventions have been created to support individuals with specific types of SUD such as alcohol use disorders or nicotine use disorders [1]. Most studies reported that digital interventions were efficacious in reducing substance use and promoting recovery efforts [1,5,6]. However, very little is known about the types, purposes, and effects of digital technology being used to support drug-related treatment and recovery for pregnant and early parenting women (PEPW) with SUD and their families. For the purposes of this scoping review, PEPW are defined as women currently pregnant and women who are parenting newborns (postpartum period) and toddlers up to four years of age. PEPW have additional sources of social and environmental stressors that complicate access to and retention in SUD treatment, thus negatively impacting treatment retention and long-term recovery efforts [7]. The current scoping review addressed this gap by exploring how digital technology has been leveraged to provide recovery support to PEPW and their families who are attempting to recover from SUDs. We evaluated the recent and limited body of research on digital interventions designed to support recovery from harmful substance use within a scoping framework [8]. Research and practice considerations regarding the feasibility of using digital technology to provide accessible resources to support parenting, self-care, and recovery from addictive substances for PEPW are discussed.

## 2. Background and Significance

Women of childbearing age are at increased risk for SUD which can have a detrimental impact on pregnancy and women and children’s health [9,10]. Although exact numbers of SUD among PEPW are unavailable globally due to reporting challenges, 2019 estimates indicate that approximately 19.5 million women (15.4%) 18 or older in the United States (U.S.) used illicit drugs in the prior year [11]. Moreover, the number of women with opioid use disorder at the point of infant delivery quadrupled from 1999–2014 [12]. These recent numbers do not account for harmful licit drug use during pregnancy which can also worsen maternal and pregnancy outcomes. The most frequently used substances in pregnancy are tobacco, alcohol, cannabis, opioids, and other illicit substances [13]. According to Washio et al. (2022) [14], polysubstance use in pregnancy is common, as well as co-occurring mental health disorders, interpersonal and environmental challenges, and limited and disrupted access to prenatal care, all of which may compound adverse maternal and neonatal health outcomes. The Centers for Disease Control and Prevention (CDC) has estimated that 10% of pregnant women in the U.S. reported current alcohol use, with alcohol use being more prevalent in the first trimester than the second and third trimesters [10]. Of those with alcohol use during the pregnancy period, about 40% reported concurrent use of other substances [10]. In another report analyzing available data on opioid use in pregnant women, Harai and colleagues estimated that the rates of neonatal abstinence syndrome (NAS) and maternal opioid use disorders have significantly increased in most states in the United States [15]. Substantial evidence exists that maternal SUD often led to negative consequences for mothers, with negative acute and long-term health outcomes for their offspring including premature delivery, NAS, congenital anomalies, low birth weight, developmental delays, and infant mortality [13]. These data underscore the need to prioritize screening and increase access to treatment for PEPW with SUD to improve the health of the mother and baby during the perinatal period and beyond.

Pregnant and early parenting women attempting recovery from SUD, especially those living in rural areas, have limited access to resources to promote recovery and parenting in their communities [16,17]. These vulnerable women may find it difficult to engage in drug recovery programs geographically located in urban settings, especially PEPW who have low income, lack personal transportation, or are uninsured [17]. For these reasons, providing digital interventions that support both recovery self-management and parenting offers an attractive and accessible resource to health-related care to facilitate better health outcomes for the entire family [18].

Prior Research. To our knowledge, no scoping review to date has explored the digital interventions that specifically target PEPW with SUD. By contrast, systematic reviews and meta-analyses have examined the effectiveness of digital interventions targeting SUD for women of childbearing age [2,14] or for women generally with or without substance use [19]. Some systematic reviews focused specifically on digital interventions targeting one or more substances (e.g., nicotine, alcohol, etc.) during the pregnancy period only [6,20,21]. One review focused on e-health interventions for women in the perinatal period, but this study focused on health promotion generally and was not specific to women with SUD [22]. 

Goal of This Review. Existing digital supports specifically for PEPW recovering from SUD have been underexplored in the literature. We aimed to (1) describe the design, methods, results, and implications of each study, (2) identify how different types of digital technology have been used to support PEPW with SUD, and (3) examine the effectiveness of the digital interventions on reported maternal, neonatal, and family outcomes. 

## 3. Materials and Methods

Search Strategy. A systematic literature search to identify relevant articles focused on digital technology use for PEPW with substance use was guided by the Scoping Review Framework of Arksey and O’Malley (2005) [8]. This scoping review has no registered protocol and has been reported following the guidelines of PRISMA-ScR (Preferred Reporting Items for Systematic Reviews and Meta-analysis extension for Scoping Reviews). 

A search for relevant articles containing keywords related to pregnancy, substance use disorder, parenting women, and digital health interventions was conducted. The search was carried out independently by the first author and two graduate research assistants, all of whom were assisted by the same health sciences librarian specialist at three different time points over a 3-year period. The original search was conducted in August 2019, but there had been a significant increase in digital technology use as part of substance use treatment delivery since COVID-19, which led the authors to repeat the search in October 2021 and August 2022. Articles published through 1 August 2022, were included in the search. This review included samples of diverse types of original studies, published from 2010 to 2022 within the Cumulative Index for Nursing and Allied Health (CINAHL), PsycInfo, PubMed, and ProQuest databases. 

Search Terms. Both medical subject headings (MeSH) and free-text keywords were used to search relevant articles in these databases that were published in the English language between January 2010 and August 2022. Search term phrases such as telehealth and women in recovery, telehealth, and parental support yielded no relevant results pertaining to our research. The first author then expanded the search using broader terms such as substance use and mothers, alcohol, and drugs and early parenting, and telehealth for women in recovery. Subject headings included women in recovery, mothers in early recovery, substance use, and telehealth. MeSH phrases included: Mothers OR Mother* OR Pregnant* OR Pregnancy women [MeSH Terms] OR Postpartum OR Parenting AND (Addiction OR Chemical dependence OR Drug use OR Alcoholic drug usage) AND (Smart phones OR Smartphones OR iPhone OR Mobile technology OR Telehealth), (Mother OR Mom) AND (Opioid addiction) AND (Telehealth), (Pregnant* women) AND (Drug use) AND (Telephone intervention) AND (Telehealth), and (Pregnant women) AND (Drug dependence) AND (Telehealth OR Mobile technology). Additional search terms included Mothers in early recovery, Telehealth, Telehealth treatment for mothers, Treatment of substance use and telemedicine for mothers in early recovery, Parents and substance abuse and recovery, and Parental support and mothers in recovery and telehealth. The full search strategy is outlined in Appendix B.

Eligibility. Inclusion criteria for in-depth reviews were full-text, academic peer-reviewed journal articles that reported original research results and included an abstract and references. Articles were included if the research identified (1) a digital intervention targeting substance use during pregnancy, postpartum, or early parenting period; and (2) a substance misused such as nicotine, alcohol, or illicit drugs by PEPW. Diverse methods for original research, randomized control trials, pre- and post-nonrandomized intervention studies with no control group, pilot/feasibility studies, and observational, and qualitative studies were included. 

Systematic reviews and meta-analysis were excluded from this review. However, reference lists of studies from systematic reviews on related topics were screened and individual studies were included if inclusion/exclusion criteria were met. Book chapters, opinion papers, research protocols, case reports, theses, and unpublished doctoral works were excluded. Studies were excluded if they were not written in English, if interventions were not digital in nature, or if the population of focus did not include PEPW with a history of SUD or substance misuse. For the purposes of this review, educational videotapes and digital ultrasounds were not considered a digital intervention that met inclusion criteria. 

Final Selection Process. The first author and graduate research assistants manually screened the list of titles and abstracts of identified articles and removed duplicates and articles that were screened out based on inclusion/exclusion criteria. During screenings when graduate assistants were uncertain whether the study was targeting PEPW or whether the intervention fit within our definition for digital technology, the conflicts were resolved by the first author (PR). Full texts of articles were obtained, reviewed, and screened by the first author, resulting in 30 articles for inclusion in this review. 

Data Extraction. Extracted information included: (a) year of publication, (b) study type, (c) purpose of the study, (d) type of digital intervention used, (e) the study population of PEPW, (f) any family/external support systems included in the digital intervention, (g) the substance(s) targeted for the digital intervention, (h) a description of the interventions, and (i) key findings.

## 4. Results

Database searches along with other methods (e.g., citation searching; assistance from the health librarian specialist) yielded 4060 articles of which 3863 underwent an initial screening of title and abstract after removal of duplicates (see Figure 1—Prisma Flow Diagram for Scoping Review). In addition, the reference lists of systematic reviews and meta-analysis (n = 144) that were on a similar topic were also searched. After assessing the title and the abstract of the articles, 3794 articles were excluded, and 69 articles underwent full-text screening by the first author. Identified articles underwent title and abstract screening, and when relevant, full text review. Ultimately, 27 original studies and 30 articles met all the selection criteria and were included in this review (Figure 1). The detailed characteristics of the identified studies are included in the Appendix A.

### 4.1. Study Time Period

Twenty-four of the identified studies were published prior to the start of the COVID-19 Pandemic (i.e., 2010–2019), and 6 studies were published after the COVID-19 Pandemic started (i.e., 2020–2021). The focus of the studies during the COVID-19 pandemic shifted to necessary telehealth utilization [24,25,26] for continued health care service delivery for PEPW, texting to promote smoking cessation for pregnant women [27], smoking cessation smart phone application with financial incentives [28], and identifying online health community supports for pregnant women with harmful opioid use by exploring web-based posts [29].

### 4.2. Study Designs

A variety of study designs were used. Fourteen of the studies were randomized controlled trials (RCTs) [27,30,31,32], and one study used a randomized program evaluation design [33]. A nonrandomized control trial was reported in one study [24] and another research team used a naturalistic non-equivalent control group quasi-experimental design [34]. Researchers who conducted one study reported using an experimental evaluation design [35]. Findings from a secondary analysis of the Martino et al.’s 2018 randomized control trial were reported by Forray and colleagues (2019) [36]. Results from a program evaluation where traditional in-person care services changed to telehealth delivery were reported in two articles [25,26]. A mixed methods design [37] and mixed-methods analysis approach [29] were used by other investigators, and seven studies were described as using pilot, observational, or feasibility designs [28,38,39,40,41,42,43]. 

### 4.3. Types of Digital Technology

The two most common digital interventions in the identified studies used were mobile phones and computer platforms. Mobile phones were used to deploy text-messaging strategies to decrease substance use or promote drug abstinence [25,27,31,32,33,43,44,45]. Mobile phones were also used for telephone counseling [46,47], as well as an e-learning smoking cessation program accessible via smartphone with Internet access [35]. Digital technology supported by computer-based platforms was equally used as an intervention strategy. Computerized brief interventions for substance use treatment and referral if indicated were the foci in most studies that used computer-based platforms [30,36,37,48,49,50,51]. Other study investigators reported using computer-based platforms to deliver a digital intervention focused on healthy pregnancy domains including smoking cessation [38] and computer-tailored feedback letters to address maternal alcohol use based on participant characteristics [52]. Four of the interventions in the selected studies were web-based platforms where the digital intervention was accessible online [25,29,46,53]. Two forms of digital technology were used for brief online or text-message based screenings for mental health and SUD referrals in response to COVID-19, when treatment programs were converted to online platforms [25,26]. Harris and Reynolds (2015) used telephone counseling combined with a web-based contingency management program [46].

### 4.4. Target Population to Include Family/External Support Systems

The reported age range for participants across all studies was 16–56 years. Participants’ median ages ranged from 23.2–34.6 years. Three studies did not provide information regarding the age of the sample. Only four studies were conducted outside of the U.S. The other studies were conducted in England, Japan, or the Netherlands. Researchers in most studies did not report whether they recruited from rural, urban, or suburban communities, although in several studies, recruiting underrepresented minorities was highlighted (see Appendix A). Researchers, in two studies, targeted rural populations whereas in two other studies, urban populations within inner city neighborhoods were recruited. The overwhelming majority of studies (n = 26; 86.7%) involved pregnant women, while women in the postpartum period were enrolled in three studies. In one study, both mothers and fathers of children aged 2–12 years participated in an online parenting program [39]. No family members or other people who may have provided social support to PEPW were engaged with digital intervention in any study. However, external support systems for the targeted groups were addressed in three studies. Peer support leaders with a lived experience of perinatal SUD were recruited to facilitate a digital storytelling intervention for pregnant and early postpartum women (i.e., up to eight weeks after delivery) [40,41]. In the Johnston et al., 2019 study, a smart phone application was used to provide recovery support for women mandated into drug treatment. Although the smart phone application was specifically for the women with SUD, the intensive treatment attended by participants provided additional supports. These supports included individual and group counseling services, psychiatric services, case management, family counseling and education, behavioral health education, and assistance establishing a support network with peers, providers, and sober communities [34].

### 4.5. Summary and Synthesis of Results

Substances targeted for the digital intervention were classified into four categories based on the drugs used by the women in each study: (1) polysubstance, (2) nicotine/smoking, (3) alcohol, and (4) illicit drugs. The categories are summarized from the greatest to the fewest number of studies in a drug category. 

#### 4.5.1. Polysubstance

Pregnant women were included as participants in nine out of the twelve studies in the category of polysubstance use [26,31,34,36,40,41,42,50,51], whereas three of the twelve studies in the category of polysubstance use enrolled early postpartum women [48,49] or early parenting women that included women with a child up to four years old [39]. In five studies, computer platforms were used to deliver computer-assisted drug screening, brief intervention, and referral to treatment sessions to decrease harmful substance use through motivational interviewing techniques [36,48,49,50,51]. One pilot study, reported in two articles, used a digital storytelling intervention on peer recovery support specialists with a lived experience of perinatal SUD [40,41]. In two studies, mobile apps were used to deliver parenting and recovery support resources [34,39] and in one study participants completed ecological momentary assessments via a mobile application to track trauma symptoms, prenatal bonding, and substance use [42]. An evidence-based text-messaging program with underpinnings in a behavioral change theory was used in one RCT to address smoking and alcohol use in pregnant women [31]. In one program, a telemedicine platform was used to deliver telepsychology to pregnant and early postpartum women [26].

Researchers in all 12 studies collected self-report measures. In five studies, urine and/or hair samples were obtained to determine polysubstance use. A significant reduction in polysubstance use outcomes was reported in seven of the twelve studies [31,34,36,39,48,50,51]. Maternal outcomes related to polysubstance drug use and other high-risk maternal behaviors [31,51] were reported at varying time points after the intervention in most studies [31,36,48,49,50,51]. Johnston et al., 2019 reported retention in drug treatment among women with polysubstance use, and other researchers reported the feasibility, acceptability, and use patterns of the digital intervention [26,42,49,51]. Maternal and neonatal outcomes, as well as feasibility outcomes, were reported in one study [39]. The researchers found significant improvement in parenting outcomes, a reduction in child behavioral problems, and user engagement associated with the digital version of an established evidence-based parenting program (i.e., Triple P) [39]. Neither maternal nor neonatal outcomes were reported in the three studies [26,40,41,42]. 

#### 4.5.2. Nicotine Use/Smoking

All ten studies targeting nicotine use or smoking involved pregnant women as participants [27,28,35,38,43,44,45,46,47,53]. Of the six RCTs, text messaging was the most common digital intervention (n = 3) [27,44,45]. Other digital interventions in the RCTs were telephone counseling (n = 1) [47], telephone counseling with a web-based contingency management program (n = 1) [46], and a web-based platform in which education, videos, and ex-smoker testimonials were used to motivate users to quit smoking during pregnancy (n = 1) [53]. Results from three of the RCTs showed a significant decrease in nicotine use or abstinence in pregnancy [27,44,47], while the results from other studies showed some positive, but non-statistically significant trends [45,46,53]. 

Other studies that enrolled women who used nicotine or smoked included a study with an observational design [43], two pilot/feasibility studies [28,38], and an experimental evaluation study [35]. Results from three of four studies identified positive effects in smoking cessation or reduction following the digital interventions. Digital interventions included smart phone application to deliver incentives for smoking cessation [28], cessation support text-messaging with no Internet required [43], and cell phones to deliver an e-learning program with Internet required [35]. The overall feasibility and acceptability of the digital intervention that targeted several health behaviors, including smoking cessation, fruit, and vegetable consumption, and stress management using a computer platform was evaluated in one study [38]. The authors reported that women’s intentions to make health behavior changes, including smoking cessation, increased because of the digital intervention.

The digital interventions used to target nicotine use/smoking in the identified studies had underpinnings in behavior change theories and evidence-based practice guidelines. The targeted mechanism of action for most of these digital interventions was to improve women’s self-efficacy and self-management for smoking cessation. Biochemical analyses to objectively determine smoking cessation through saliva, urine, or hair testing to measure women’s cotinine levels or carbon monoxide exhalations were used in seven studies. Cotinine cutoff points for those using saliva or urine sampling were inconsistent among studies, ranging from 4–10 ng/mL to determine abstinence or reduction in nicotine. However, studies within this category are unique in reporting biochemical or objective data on substance use following the digital intervention.

#### 4.5.3. Alcohol Use

There were five studies focused on alcohol use and these studies all included pregnant women as study participants [30,32,33,37,52]. Two of the five studies were RCTs (one pilot RCT and one full scale RCT) and mobile phones were used to deliver a text messaging intervention to promote health behavior change [32,33]. A computer platform was used in one RCT to deliver a brief intervention [30]. In another RCT, computer-tailored feedback letters about harmful alcohol consumption and abstinence support were delivered as part of a brief intervention to promote alcohol abstinence during pregnancy [52]. Both RCTs that used computer platforms showed significant reductions in self-reported alcohol use or abstinence. In addition, Tzilos and colleagues (2011) reported that their brief intervention resulted in higher birth weights for infants born to women in the intervention group as compared to the control group. Participants also provided high ratings for the intervention acceptability [30]. Participants who received health promoting text messages reported decreased alcohol use, particularly in the postpartum period [33]. In the other study targeting alcohol use, researchers employed a mixed methods design to evaluate the feasibility of a digital brief intervention delivered via a computer platform [37]. Primary feasibility outcomes showed high intervention acceptability [37].

The digital interventions used to target alcohol in the identified studies also had underpinnings in behavior change theories. Although the study outcome was centered around alcohol use patterns, researchers who used text-messaging digital interventions addressed healthy eating and nicotine use as well [30,47]. The hypothesized mechanisms of action for the digital interventions were tailored education about harmful alcohol use during pregnancy coupled with motivational interviewing techniques [28,34,49], self-efficacy enhancement [30,47,49], and improved self-management [34,49] to decrease alcohol consumption. Primary health outcomes focused on reducing alcohol use or abstinence [30,47], acceptability of the digital intervention [34], and birth outcome data taken from medical records [28]. None of the investigators used biochemical analysis to determine alcohol use and effects of the intervention were not measured beyond the early postpartum period. 

#### 4.5.4. Illicit Drugs

Three studies primarily focused on opioid misuse and targeted pregnant women [24,29] or PEPW [25]. Liang et al., 2021 used a mixed methods analysis approach involving inductive thematic analysis and computational analysis of theme-related posts in a public online health community to identify characteristics of women who self-manage their opioid use during pregnancy and to identify women’s self-management support needs [29]. Among the pregnant population accessing an online health community for support, self-managed withdrawal was statistically more common than professional treatment. Five self-management support need themes emerged for PEPW with opioid disorders, including the potential adverse effects of gestational opioid use, protocols for self-managed withdrawal, safe pain management during pregnancy, hospital policies and legal procedures related to child protection/custody if opioid use were detected/reported, and strategies for navigating external in-person support systems [29]. In a nonrandomized, two-group study, researchers compared in-person treatment for opioid use disorder to telemedicine-delivered substance use treatment among women who were pregnant [24]. Results indicated no significant differences in treatment retention rates or neonatal abstinence syndrome between the two modes of treatment delivery [24]. In the last study, the research team used a web-based platform to provide online screenings for PEPW with opioid use disorders [25]. Online screening showed a positive and significant uptake in utilization of maternal mental health and substance use disorder screenings and treatment services [25]. 

## 5. Discussion

The purpose of this scoping review was to determine how digital technology has been leveraged to support PEPW’s substance use recovery. Studies were conducted in several different countries and with diverse samples of PEPW. Varying study designs were used, and several researchers reported primarily on the feasibility and acceptability of the digital intervention indicating an early stage in the science of supporting PEPW with digital interventions. Digital interventions tested in the studies primarily targeted women with polysubstance use disorders (40.0%) or nicotine use/smoking (33.3%). Fewer researchers evaluated digital approaches targeting PEPW with alcohol use (16.7%) or illicit drug use/opioid use (10.0%). Many of the nicotine use/smoking cessation intervention studies were unique in reporting gold standard objective substance use outcomes, but there was substantial variety in the thresholds employed. The overwhelming majority of the studies focused on the digital interventions in pregnant women (89.7%), suggesting a dearth of research on how digital technologies may support postpartum and early parenting women with SUD. Many researchers reported on some aspect of maternal health outcomes, but only three studies included neonatal (n = 2) or child (n = 1) health outcomes. 

Addressing substance misuse within the context of the PEPW’s lived experience in natural communities is essential in developing meaningful and long-lasting interventions that promote positive recovery and parenting outcomes [54]. One way to do this is by creating community-based partnerships with parenting women in recovery from SUD to develop or craft a tailored digital recovery support intervention specific to their unique contextual needs [18,54,55]. None of the researchers reported using a community-based participatory approach to develop or tailor the digital intervention. 

Additionally, none of the researchers directly targeted family or external support systems to engage in the intervention alongside PEPW, although in one study the digital intervention was offered to mothers or fathers for independent use [39]. Further, in the reviewed studies, most of the digital interventions were offered during pregnancy until eight weeks after the birth. No additional tailored education or supports were reported to address the ongoing mental health, self-care, and community resources PEPW with SUD may need in their natural living environments during periods of increased susceptibility and risk exposure, particularly during the entire first year of the baby’s life (i.e., fourth trimester) when they are at significantly greater risk for relapse, and fatal overdose [55]. 

As substance use is often common among partners, family members, or support persons [56,57], it is possible that interventions that engage PEPW’s support networks may be more effective and digital technology may be a promising intervention strategy [58]. Digitally enhanced family focused psychosocial interventions have been pilot-tested for adolescent populations with both mood disorders and chronic pain with significant improvement in adolescent health at the end of the intervention period [59,60]. Further, non-digital approaches that engage a broader support network for PEPWs have been shown to promote substantial improvements in substance use [61,62] and thus expanding this approach into a digital delivery format holds promise for improving outcomes by increasing sustained access. Future research with digital interventions that include family and recovery supports are needed to promote PEPW’s recovery efforts as they transition back into their natural living environments. Further research is necessary to understand how family-based engagement and enhancement approaches in digital health interventions can improve maternal and child health outcomes over time.

Mobile phones (30.0%) and computer-based platforms (30.0%) were the most reported digital approaches, followed by mobile applications (13.3%), web-based platforms (6.7%), telemedicine platforms (6.7%), digital storytelling (6.7%), and a combination of mobile and computer or web-based platforms (6.7%). Although digital platforms have been used extensively over the last couple of years to promote accessibility to care for individuals with SUD, most of the studies targeting PEPW identified for this review were conducted prior to the start of the COVID-19 pandemic. 

Noteworthy findings were the successful use of mobile phones requiring only text-based messaging (e.g., no Internet capability) to support PEPW with harmful substance use. Even if more advanced digital approaches are not feasible in low resource countries, text-based technologies using mobile phones that do not require Internet capability can have a significant role to play in reducing harmful substance use as they were also shown to be effective. Moreover, technology-based interventions, particularly those delivering screening and brief interventions via computer platforms [63] and digital designs based on underpinnings of a well-established parenting program structured to provide education and skills training (i.e., Triple P), were effective in reducing self-reported polysubstance misuse. These findings were consistent with the existing literature [63,64]. Moreover, notable were positive findings in length of stay and retention in treatment from a mobile app delivering recovery support (i.e., A-CHESS) for pregnant women recruited from an intensive SUD treatment center. Overall, most technology-based interventions were effective, but to varying degrees, in reducing nicotine use/smoking cessation. The digital interventions targeting alcohol use via computer platforms and text-messaging were effective in reducing self-reported alcohol consumption. Digital interventions targeting illicit drug, primarily opioid misuse, were less common and more preliminary, reporting feasibility outcomes of telemedicine platforms for service delivery and an analysis of an online health community. More research is needed to determine the benefits of leveraging digital technology to support drug treatment, recovery, and family outcomes especially for PEPW with opioid use disorders. 

Research and Practice Implications. Limited access to SUD treatment and services for PEPW can exacerbate drug use. Digital technology has been utilized more frequently since the COVID-19 pandemic to improve health care access and availability to drug treatment generally. However, less attention has been given to PEPW outside of telemedicine platforms to deliver medication-assisted treatments. Intentional use of community stakeholder involvement to develop and contextualize the digital psychosocial support intervention may be useful to increase utilization, sustain engagement for parenting and recovery supports, and promote positive maternal health outcomes. No identified studies targeted PEPW as community stakeholders in digital intervention development, thus warranting future work in this area. In addition, other digital technology applications, such as smartwatches, robots, virtual assistant devices, and home smart speakers, may have potential benefits for delivering self-management strategies and supports for PEPW with SUD, but have not yet been studied. Identifying effective ways to support PEPW through the pregnancy period and throughout the first years of the child’s life utilizing digital technology is essential in promoting long-term recovery and well-being for the entire family.

Some specific limitations of the current scoping review should be noted. First, the literature search for this review was limited to published studies after the year 2010 and studies published in English. Further, the search terms and phrases used for determining relevant studies may have contributed to an exclusion of other clinically relevant articles applicable to PEPW, substance use, and digital technology interventions.

## 6. Conclusions

In summary, 27 studies (30 articles) met the inclusion and exclusion criteria for this scoping review. Digital interventions targeting PEPW with polysubstance use and nicotine use/smoking cessation disorders were more frequently reported than studies addressing other substances. Most digital interventions showed positive effects to varying degrees in reducing substance use in PEPW. In contrast, not all interventions had significant effects in reducing substance use and in some studies the digital technology was not an intervention, but rather a mechanism to explore substance use severity with other associated mental health symptoms (i.e., ecological momentary assessments), or assess the feasibility of employing the digital technology as a screening and treatment platform for service delivery (i.e., telemedicine, telepsychology). Future work should determine the benefits of leveraging different types of digital technology to support drug treatment, recovery, and family outcomes specifically for postpartum and early parenting women with opioid use as there were very few studies that targeted this important and growing population. In addition, more research is needed to explore the effective ways to utilize digital technology that engage the entire family in the intervention. When PEPW live within the context of family members who are actively using drugs, it can undermine recovery efforts leading to harmful effects for the mother and the baby. Effective digital interventions that provide recovery and parenting support for PEPW will need to address harmful substance use within the context of women’s lived experiences and social relationships in their natural communities. 

## Figures and Tables

**Figure 1 ijerph-20-04457-f001:**
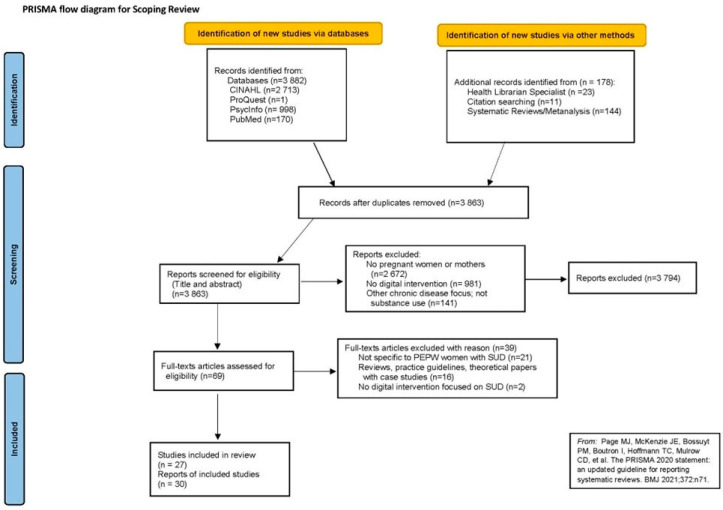
Prisma diagram for scoping review [23].

## Data Availability

The data presented in this study are available on request from the corresponding author.

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
