# Peer review of "Leveraging Digital Technology to Support Pregnant and Early Parenting Women in Recovery from Addictive Substances: A Scoping Review"

_ijerph, 2023, doi:10.3390/ijerph20054457_

Round 1

Reviewer 1 Report

Overall this is a very interesting topic that is relevant in the current time.

Line 226 - "into" is not the best wording

Substance use/polysubstance use: the authors did not identify whether the use was based on self report or testing.

Overall: there is a good amount of redundancy between sections and the paper can be written more concisely.

The topic that is being discussed is very interesting, however, looking at pregnant and post partum women is combining two potentially very different population of patients.  Also, because the outcomes are so different in all the studies, it is really difficult to extrapolate any significant result from the discussion.

It may be beneficial to put some of these results into a table or graph form as it is very difficult to follow in its current form.

Author Response

Thank you so much for this excellent feedback detailing suggested revisions to our manuscript. We have thoroughly addressed each point and submitted a Table detailing the address for each item referenced.

See below:

Reviewer Comments (RC):

Author revisions based on feedback

RC 1:

Overall this is a very interesting topic that is relevant in the current time.

Thank you. No revisions needed.

Line 226 - "into" is not the best wording

We agree. Thank you for this excellent feedback. We have changed the word “into” to “with” for greater sentence clarity. The sentence now reads much clearer as “No family members or other people who may have provided social support to PEPW were engaged with the digital intervention.”

Substance use/polysubstance use: the authors did not identify whether the use was based on self-report or testing.

Thank you for this feedback. On page 6, line 271, we have added “Researchers in all 12 studies collected self-report measures. In five studies, urine and/or hair samples were obtained to determine polysubstance use.”

Overall: there is a good amount of redundancy between sections and the paper can be written more concisely.

Yes, you are correct and thank you for this wonderful feedback. We have carefully reviewed the manuscript again and have removed redundancy in all sections of the paper for greater clarity and flow.

The topic that is being discussed is very interesting, however, looking at pregnant and post-partum women is combining two potentially very different population of patients. Also, because the outcomes are so different in all the studies, it is really difficult to extrapolate any significant result from the discussion.

Thank you for bringing this to our attention. We have provided more detail regarding the participants characteristics across studies under the target population section (lines 226-235). In addition, we provided clear wording to make the relevant findings stand out in the results section. We believe this makes it easier to extrapolate significant findings from both the Results and Discussion section.

It may be beneficial to put some of these results into a table or graph form as it is very difficult to follow in its current form.

Thank you for this feedback. We have carefully considered the best approach to address this challenge, and have done this by strengthening the summary statements in each subcategory of our results sections where we have highlighted relevant findings. As the general purpose of the scoping review was to provide an overview of how digital technology has been employed to support this vulnerable population, we believe that showing the variability in digital technology use and types for PEPW aligns directly with the aim of the scoping review. We have also included detailed summaries of the selected articles in Supplemental Table 1. If the reviewer has a specific recommendation about data that should be highlighted more and separately, we’d be more than happy to do that.

Reviewer 2 Report

This is a well-written manuscript that highlights an important issue among pregnant women and early parenting women. There are some long sentences that could be broken up to improve the flow of reading. I have some more specific comments below for the authors' consideration.

Introduction

·        Can the authors clarify whether the CDC’s estimate of 10% of pregnant women who reported alcohol use was based on national data or something else?

·        Can the authors include a few lines to indicate how big of an issue SUD is within this population? For instance, what percentage of US-based PEPW use opioids? It is not immediately apparent if the focus of the manuscript is domestic or international.

Results

·        Page 6, lines 261-277: This paragraph seems repetitive. The sentence that starts with “Paterno and colleagues” is confusing. The first sentence of this paragraph may suffice if the authors simply list the three studies at the end.

·        What was the age range of women across the studies?

·        Where the studies conducted? Even one sentence on this would be helpful.

·        Also, can the authors include a breakdown of studies by urban vs. rural populations, given some of the details they shared about these groups earlier in the manuscript?

Discussion

The first part of this section seems to be from the journal’s instructions. The authors should delete it.

Author Response

Thank you so much for taking the time to render such a thoughtful excellent review. We have tried to address each item separately and have included a Table below to provide details on how we have revised the manuscript based on your feedback. We believe the manuscript is clearer and much stronger because of your suggested revisions and for this, we are grateful.

See below:

Reviewer 2 Comments with Authors Revisions

RC2: (Introduction)

This is a well-written manuscript that highlights an important issue among pregnant women and early parenting women. There are some long sentences that could be broken up to improve the flow of reading. I have some more specific comments below for the authors' consideration.

Thank you for this feedback. We have critically edited the manuscript again to capture and correct long and complicated sentences throughout the manuscript. In places where this occurred, we have broken up long sentences to improve reading flow.

Can the authors clarify whether the CDC’s estimate of 10% of pregnant women who reported alcohol use was based on national data or something else?

Thank you for this feedback. We have clarified that this estimate is based on national data. See revised statement below: “The Centers for Disease Control and Prevention (CDC) has estimated that 10% of pregnant women in the U.S. reported current alcohol use, with alcohol use being more prevalent in the first trimester than the second and third trimesters [10].”

Can the authors include a few lines to indicate how big of an issue SUD is within this population? For instance, what percentage of US-based PEPW use opioids? It is not immediately apparent if the focus of the manuscript is domestic or international.

Yes of course. On Page 2, line 67-73 has been added. “Although exact numbers of SUD among PEPW are unavailable globally due to reporting challenges, 2019 estimates indicate that approximately 19.5 million women (15.4%) 18 or older in the United States (U.S.) used illicit drugs in the prior year [11]. Moreover, the number of women with opioid use disorder at the point of infant delivery quadrupled from 1999 -2014 [12]. These recent numbers do not account for harmful licit drug use during pregnancy which can also worsen maternal and pregnancy outcomes.”

RC2: (Results)

Page 6, lines 261-277: This paragraph seems repetitive. The sentence that starts with “Paterno and colleagues” is confusing. The first sentence of this paragraph may suffice if the authors simply list the three studies at the end.

Thank you for this wonderful feedback. We have removed that wording after the first sentence, and then included the reference numbers for the three studies at the end of the sentence. We added this sentence to the preceding paragraph. (line 282-283)

What was the age range of women across the studies?

The age ranges and median ages for study participants have been included in the manuscript within the results section (226-234).

Where the studies conducted? Even one sentence on this would be helpful.

The geographical locations for the study have been included in the manuscript in the results section detailing characteristics of the target population (line 226-234).

Also, can the authors include a breakdown of studies by urban vs. rural populations, given some of the details they shared about these groups earlier in the manuscript?

Thank you for this suggestion. We have provided information regarding the targeted populations (i.e., urban, rural, suburban, etc.) when specifically referenced by the authors of the selected studies (lines 230-234).

RC2: (Discussion)

The first part of this section seems to be from the journal’s instructions. The authors should delete it.

Thank you for this feedback. The first part of this section did come from the journal’s instructions and was accidentally included in the manuscript. The authors have deleted it.
